# Application of Electrospun Nanofiber Membrane in the Treatment of Diabetic Wounds

**DOI:** 10.3390/pharmaceutics14010006

**Published:** 2021-12-21

**Authors:** Zhaoju Gao, Qiuxiang Wang, Qingqiang Yao, Pingping Zhang

**Affiliations:** School of Pharmacy and Pharmaceutical Sciences, Shandong First Medical University & Shandong Academy of Medical Sciences, Jinan 250000, China; zhaojugao6@sina.com (Z.G.); qiuxiangwangoo@sina.com (Q.W.)

**Keywords:** electrospun nanofibers, diabetic wounds, polymer electrospun fibers, nanoparticles, drugs, cell

## Abstract

Diabetic wounds are complications of diabetes which are caused by skin dystrophy because of local ischemia and hypoxia. Diabetes causes wounds in a pathological state of inflammation, resulting in delayed wound healing. The structure of electrospun nanofibers is similar to that of the extracellular matrix (ECM), which is conducive to the attachment, growth, and migration of fibroblasts, thus favoring the formation of new skin tissue at the wound. The composition and size of electrospun nanofiber membranes can be easily adjusted, and the controlled release of loaded drugs can be realized by regulating the fiber structure. The porous structure of the fiber membrane is beneficial to gas exchange and exudate absorption at the wound, and the fiber surface can be easily modified to give it function. Electrospun fibers can be used as wound dressing and have great application potential in the treatment of diabetic wounds. In this study, the applications of polymer electrospun fibers, nanoparticle-loaded electrospun fibers, drug-loaded electrospun fibers, and cell-loaded electrospun fibers, in the treatment of diabetic wounds were reviewed, and provide new ideas for the effective treatment of diabetic wounds.

## 1. Introduction

Wounds are caused by destruction of the integrity of skin tissue, which is often accompanied by a loss of organismal material. Wounds can be divided into acute and chronic wounds [1,2], superficial, deep, and whole cortical wounds [3,4,5], clean, contaminated, and infected wounds [6,7,8], abrasions, incisions, tears, ulcers, and detachment [9,10,11,12,13], according to the length of the healing time, the depth, the infection, and the injury type of the wounds, respectively.

### 1.1. Normal Wound Healing Process

Normal wound healing can be divided into four stages: hemostasis, inflammation, proliferation, and remodeling. There are complex and dynamic interactions among the four stages [14,15,16]. The wound-healing process is shown in Figure 1.

#### 1.1.1. Hemostasis Stage

The hemostasis stage of wound healing refers to the body first promoting the contraction of vascular smooth muscle cells through the neural reflex mechanism, which causes rapid contraction of the damaged blood vessels and triggers hemostasis [18]. Then platelets aggregate to form blood clots at the wound and start hemostasis [19]. Finally, platelets rupture and release growth factors (such as platelet-derived growth factor (PDGF), transforming growth factor-β (TGF-β), and epidermal growth factor (EGF)), thus attracting neutrophils, macrophages, and fibroblasts, which play a role in the subsequent healing phases [20,21,22,23].

#### 1.1.2. Inflammation Stage

The inflammation stage is the process of protecting the wound from infection by microorganisms by forming an immune barrier. The early inflammatory stage occurs 24–36 h after injury, with neutrophils playing a major role. Neutrophils engulf microorganisms and foreign bodies and then are squeezed to the surface of the wound, after which they are swallowed by macrophages [24]. The later inflammatory stage occurs 36–72 h after injury, and macrophages play a role [25]. Macrophages phagocytose microorganisms, necrotic tissue, and fragments, and secrete a large number of growth factors (such as PDGF, interleukin-1 (IL-1), and tumor necrosis factor (TNF)). Macrophages can also promote granulation tissue formation by activating keratinocytes, fibroblasts, and endothelial cells [26,27].

#### 1.1.3. Proliferation Stage

The proliferation stage mainly refers to the process of granulation tissue formation, angiogenesis, and re-epithelialization [28,29]. Under the action of growth factors produced by platelets and macrophages, fibroblasts proliferate and secrete collagen, and increase the amount of ECM components in the wound, thus promoting granulation tissue formation [30]. Angiogenesis is the key process of wound healing. The anoxic environment of wound tissue and growth factors secreted by macrophages can stimulate the proliferation of endothelial cells. Under the action of angiogenic factors, such as the vascular endothelial growth factor (VEGF) and basic fibroblast growth factor (bFGF), endothelial cells promote the formation of new blood vessels [31]. At 48–72 h after injury, keratinocytes in the wound proliferate under the action of EGF, TGF-β, and other cytokines, which stimulate re-epithelialization [32,33].

#### 1.1.4. Remodeling Stage

The remodeling stage is the final stage of wound healing. Under the action of cytokines, fibroblasts increase expression of α-smooth muscle actin (α-SMA) and transform into myofibroblasts. Under the action of myofibroblasts, the wound continuously contracts to form a scar, cell apoptosis and cell regeneration reach a balance, and components such as proteins and collagen in the ECM tend to be stable. This process takes months or even years to complete the remodeling phase [34].

### 1.2. Diabetic Wounds

A diabetic wound is simultaneously a chronic wound, deep wound, infected wound, and ulcer wound, which is characterized by high levels of inflammatory cytokines, matrix metalloproteinases (MMPs), and reactive oxygen species (ROS), difficult angiogenesis, and persistent infection [35,36,37,38].

Local ischemia and hypoxia lead to skin dystrophy, and skin dystrophy causes diabetic wounds [39,40]. The hemostatic stage and inflammatory stage of diabetic wound healing follow normal wound healing. However, some internal factors (such as vascular disease and neuropathy caused by diabetes) and external factors (such as persistent wound infection) make it difficult for diabetic wounds to transition from the inflammation stage to the proliferation stage, resulting in slow healing [41].

Complications of diabetic vascular disease lead to lack of necessary oxygen and nutrients for wound healing, which makes angiogenesis difficult and hinders the transition of wound healing from the inflammation stage to the proliferation stage [42,43,44,45]. In the later inflammation stage of normal wounds, macrophages change from pro-inflammatory M1 macrophages to repair-promoting M2 macrophages, which accelerates the process of inflammation. However, it is difficult for macrophages to change from M1 to M2 in diabetic wounds, which leads to the production of pigment epithelium-derived factor (PEDF) and inhibits angiogenesis.

The hyperglycemia environment of diabetic wounds is conducive to bacterial proliferation, and diabetic wounds have the problem of repeated bacterial infection for a long time. Diabetic patients are in a state of hyperglycemia for a long time, and their immune function is abnormal. When bacterial infection occurs in the wound, abnormal immune function often leads to an excessive inflammatory reaction, which keeps the wound in the inflammation stage for a long time [46,47,48,49].

Diabetic wounds are in the stage of inflammation for a long time, which causes neutrophils and macrophages to continuously produce a large number of inflammatory cytokines and ROS, further damaging normal tissues and cells in the wound [50,51,52]. At the same time, the presence of a large amount of ROS will cause fibroblasts to lose their normal function and slow down the deposition of ECM. The persistent inflammation stage of diabetic wounds also leads to overexpression of MMP-2 and MMP-9, which leads to the rapid degradation of ECM. The slow deposition and rapid degradation of ECM affect the adhesion of fibroblasts, resulting in slow wound healing. In addition, the expression of TGF-β and other growth factors in diabetic wounds also significantly decreases because of the long-term inflammation stage of diabetic wounds, which hinders the proliferation and migration of keratinocytes and slows down the process of re-epithelialization.

## 2. Electrospinning

Electrospinning is a technology for the preparation of nanofibers. The morphology and mechanical properties of electrospun fibers can be adjusted by properties of the polymer solution (such as the relative molecular mass of the polymer, solution concentration and viscosity, and solvent properties), process parameters (such as applied voltage, solution injection speed, and receiving distance of the fiber), and environmental conditions.

### 2.1. Critical Parameters of Electrospinning Process

#### 2.1.1. Properties of Polymer Solution

(1)Relative Molecular Mass of Polymer

The relative molecular mass of the polymer is an important parameter affecting electrospinning, which directly affects the rheological and electrical properties of the electrospinning solution. Electrospinning can be carried out only when the relative molecular mass of the polymer reaches a certain value. Polymers with low molecular weight tend to form beads during the electrospinning process, while long and continuous nanofibers can be prepared by electrospinning polymers with high molecular weight. The larger the molecular weight of the polymer, the larger the fiber diameter [53].

(2)Solution Concentration and Viscosity

The viscosity of the polymer solution can affect the morphology of electrospun fibers. In the process of electrospinning, a solution with low viscosity can readily form beads, and the increase of solution viscosity is conducive to the formation of nanofibers [54]. The entanglement of polymer molecular chains in the solution enables the solution to reach a certain viscosity. The larger the molecular weight of the polymer, the more easily chains entangle, and the higher the viscosity of the solution. When the molecular weight of the polymer is fixed, the concentration of the solution becomes an important factor affecting the entanglement of polymer molecular chains in the solution—as the concentration of the solution increases, the viscosity of the solution increases.

(3)Solvent Properties

Solvent properties have an important effect on the electrospinning process. Firstly, the solvent should have good solubility for the electrospun polymer. In addition, it should have good volatility. In the process of electrospinning, solvents that volatilize quickly can ensure the continuity of the electrospinning fiber.

#### 2.1.2. Process Parameters

(1)Applied Voltage

In the process of electrospinning, the voltage applied to the polymer fluid must exceed a certain critical value to generate sufficient electrostatic repulsion to overcome its surface tension, resulting in the formation of tiny jets to form fibers. The diameter of the fibers prepared at higher voltage is smaller, but too high a voltage will lead to increase in bead defects and fiber diameter. The type of voltage (direct current or alternating current) will also have an impact on electrospinning. When using direct current, the jet is unstable and the fiber is difficult to deposit on the receiving device, while alternating current can reduce the instability of the jet and the diameter of the fiber is smaller [55].

(2)Solution Injection Speed

Increasing the solution injection speed will lead to the expansion of the jet diameter, which increases the fiber diameter, while too slow a solution injection speed can prevent fibers forming or lead to fiber discontinuity.

(3)The Receiving Distance of the Fiber

The receiving distance of the fiber will affect the volatilization of the solvent, which directly affects the diameter and morphology of the electrospun fiber. When the receiving distance is small, solvent volatilization is not complete, resulting in uneven diameters of fibers. When the receiving distance increases to a certain extent, electrospun fibers with small and uniform diameters can be obtained. When the receiving distance is large, the electric field intensity will decrease with increase in receiving distance, which will reduce the jet velocity and weaken the tensile effect, resulting in an increase in fiber diameter. In addition, if the receiving distance is too close or too far, it will lead to the formation of beads.

#### 2.1.3. Environmental Conditions

The temperature and humidity of electrospinning will affect the deposition behavior of fibers on the collector. Lower temperature will reduce the rate of solvent volatilization, leading to the incomplete solidification of fibers. Increasing the temperature could accelerate the evaporation of solvent and produce continuous fiber, but too high a temperature will block the spinneret. The environmental humidity of the electrospinning process should be suitable; lower humidity leads to the formation of bead defects in electrospun fibers, and the fiber surface tends to adopt a porous structure.

The properties of the polymer solution, process parameters and environmental conditions, together affect the electrospinning process. In the process of electrospinning, the solvents used in some electrospinning systems were toxic solvents with poor environmental friendliness. It is necessary to consider the receiving distance when adjusting the electrospinning applied voltage. Some humidity-sensitive electrospinning systems have strict requirements on humidity. It is necessary to comprehensively consider the influence of electrospinning parameters on the electrospinning process to obtain environmentally friendly electrospun fibers with excellent properties.

### 2.2. Advantages of Electrospun Nanofiber Membranes in the Treatment of Diabetic Wounds

At present, dry gauze and hydrophilic gel are mainly used to treat diabetic wounds. Dry gauze is a common wound dressing. However, its high absorption capacity of tissue fluid can easily lead to wound dehydration, which is not conducive to wound healing. When the gauze is removed, it can also easily cause damage to the newly formed skin. Hydrophilic gels can keep the wound in a moist environment, and their structure is beneficial to gas exchange and exudate absorption at the wound site. However, hydrophilic gel materials cannot simulate the natural ECM structure, which is not conducive to the attachment, growth, and migration of fibroblasts and affects the wound-healing process.

Nanofiber membranes are one type of nanostructured material prepared by electro-spinning technology. Electrospinning nanofiber membrane refers to nanofibers in aggregate, where electrospinning nanofibers with a diameter below 1000 nm interconnect with each other to form a web structure. Electrospinning nanofiber membrane has the characteristics of large specific surface area, high porosity, small pore size, and adjustable composition. Electrospun nanofiber membranes have the advantages of adjustable composition, structure, and size. The structure of electrospun nanofibers is similar to the structure of ECM, and its porous structure is conducive to gas exchange and exudate absorption at the wound site, which leads to the regeneration of skin tissue in the wound area. Electrospun fiber membranes can also load therapeutic drugs or active ingredients and achieve controlled and sustained release of drugs through structural adjustment, which offers great prospects for application in the treatment of diabetic wounds.

### 2.3. Preparation Methods of Electrospun Nanofiber Membranes

#### 2.3.1. Uniaxial Electrospinning

Uniaxial electrospinning refers to the preparation of electrospun nanofibers by spinning the solution with a single nozzle. The operation of uniaxial electrospinning is simple, and the compounding of many active components can be achieved by adjusting the composition distribution ratio [56]. Xie added vascular endothelial growth factor (VEGF) and platelet-derived growth factor (PDGF) to polylactic-glycolic acid (PLGA) solution, prepared VEGF- and PDGF-loaded PLGA nanoparticles by compound emulsion technology, and then dispersed the nanoparticles in a mixed solution of chitosan (CS) and polyethylene oxide (PEO) to prepare the electrospinning solution. VEGF- and PDGF-loaded PLGA/CS/PEO nanofiber films were prepared by uniaxial electrospinning, with fiber diameters ranging from 130 nm to 150 nm. The fiber membrane can continuously release two kinds of growth factors within 7 d, which can significantly improve the wound healing effect in diabetic rats [57]. Uniaxial electrospinning requires that the polymers and active ingredients can dissolve in the same solvent, so the selection of solvent is more stringent. In addition, uniaxial electrospinning tends to lead to uneven distribution of drugs in the fiber, and sudden release of drugs always occurs, so stable active ingredients with high toxicity threshold are generally selected.

#### 2.3.2. Emulsion Electrospinning

Polymer and active ingredients which are difficult to co-dissolve, as well as active ingredients which are susceptible to inactivation, can be treated by emulsion electrospinning to prepare drug-loaded fibers. Emulsion electrospinning refers to the method of first mixing the aqueous phase and organic phase to form emulsions, and then electrospinning the emulsions to prepare nanofibers with a core/shell structure [58]. By using emulsion electrospinning, active components which can easily be deactivated and unstable can be loaded into the fiber core layer to improve their stability. Raghunath added easily oxidized vitamin C, easily deactivated EGF, and insulin into a PLGA and collagen mixed solution to make an emulsion. PLGA/collagen nanofiber membranes loaded with various bioactive substances were prepared by emulsion electrospinning. The fiber diameters were 210 ± 62 nm. The release of EGF from this fiber membrane in 8 h was as high as 97%, the release of insulin in 25 h was about 80%, and the release of vitamin C in 12 h was 30%. The prepared fiber membrane can simultaneously deliver a variety of bioactive substances, and the synergistic effect of various bioactive substances can promote the proliferation of keratinocytes and fibroblasts, which is conducive to diabetic wound healing [59]. The construction of a stable emulsion system is key to emulsion electrospinning. At present, the emulsion is mainly prepared by a high-energy emulsification method, which consumes much energy and may destroy the active components of drugs. The exploration of low-energy emulsification methods is currently a research focus in emulsion electrospinning.

#### 2.3.3. Coaxial Electrospinning

For easily inactivated protein drugs, coaxial electrostatic spinning can be used to protect the easily inactivated ingredient. Coaxial electrospinning refers to a method of preparing nanofibers with a core/shell structure by spinning the solution with a coaxial nozzle. Coaxial electrospinning can simultaneously spin two different component solutions, which can prevent mixing interference between different active components. The core/shell structure of coaxial electrospinning fibers can also achieve controlled drug release [60]. Lee dissolved PLGA in 1,1,1,3,3,3-hexafluoro-2-propanol (HFIP) to obtain a shell spinning solution and used insulin glargine as the nuclear layer spinning solution. Insulin-loaded core/shell nanofibrous scaffolds with insulin as the core layer and PLGA as the shell layer were prepared by coaxial electrospinning. Core/shell structure nanofibrous scaffolds can protect the activity of insulin and the fiber membrane can slowly release insulin for 28 d. In vivo experiments in Sprague–Dawley rats showed that nanofibrous scaffolds can increase expression of TGF-β in the wound and promote wound healing in diabetes mellitus [61]. Coaxial electrospinning is widely used in biomedical fields, but requires that the core and shell solutions must be solidified synchronously in the electrospinning process, which means the method has higher requirements for the preparation process [62,63,64].

## 3. Application of Electrospun Nanofiber Membranes in the Treatment of Diabetic Wounds

Polymer electrospun fibers, nanoparticle-loaded electrospun fibers, drug-loaded electrospun fibers, and cell-loaded electrospun fibers prepared by uniaxial, emulsion and coaxial electrospinning can be used to treat diabetic wounds.

### 3.1. Treatment of Diabetic Wounds with Polymer Electrospun Fibers

#### 3.1.1. Treatment of Diabetic Wounds with Electrospun Synthetic Polymer Fibers

Synthetic polymer electrospun fibers have good mechanical strength and stability, and the fibers, with a specific structure and properties, can be directly applied in the treatment of diabetic wounds. Maggay prepared an electrospinning solution by dissolving polyvinylidene fluoride (PVDF) and zwitterionic polymer poly(2-methacryloyloxyethyl phosphorylcholine-co-methacryloyloxyethyl butylurethane) (PMBU) in a mixed solvent of dimethylformamide (DMF)/acetone (*v*/*v* = 6:4), and zwitterionic PVDF membranes (P5) were constructed by uniaxial electrospinning. PMBU can improve the hydration of the P5 membrane, reduce the biological pollution of protein and bacteria, improve blood fusion, and thus promote wound healing. A 24 h bacterial adhesion experiment showed that the bacterial adhesion number of P5 was 1500 cells/mm^2^, which can effectively prevent bacterial adhesion. The diabetic wounds of mice were treated with P5, PVDF fiber membrane (P0), and the commercial wound dressing DuoDerm, respectively, and wounds treated with 3M Tegaderm Film were used as a control. After 14 d of wound treatment, the wound closure rate of the P5 group was 85%, while that of the P0 group and the DuoDerm group was 81 and 90% respectively. The wound-healing effect of the P5 group was similar to that of the DuoDerm group, and had a better diabetes wound-healing effect [65]. The photos of the treatment of diabetic wounds with the P5 membrane are shown in Figure 2.

#### 3.1.2. Natural/Synthetic Polymer Blended Electrospun Fibers in the Treatment of Diabetic Wounds

Natural polymer (such as CS, type I collagen, gelatin (Gel), and silk sericin (SS)) fibers have good biocompatibility, but the mechanical strength of natural polymer fibers is poor in the environment of body fluids, which limits the application of natural polymer fibers to some extent. Based on the advantages of easy degradation and good biocompatibility of natural polymers, blending natural polymers with synthetic polymers (such as polyvinyl alcohol (PVA) and polycaprolactone (PCL)), or modifying synthetic polymer fibers with natural polymers on their surface, can prepare natural/synthetic polymer fibers with high mechanical strength, which can improve the poor mechanical strength of natural polymer fibers while maintaining the advantages of natural polymers, and expanding the application of natural polymers.

Gholipour-Kanani prepared CS/PVA and PCL/CS/PVA electrospinning nanofiber membranes using uniaxial electrospinning, which were used to treat diabetic wounds. Diabetic rats were randomly divided into three groups: the CS/PVA fiber membrane treatment group (S1 group), the PCL/CS/PVA fiber membrane treatment group (S2 group), and the untreated diabetic wound group (the control group). The initial wound area of diabetic rats was 50.25 ± 0.01 mm^2^. After 20 d, the wounds of S1 and S2 groups were basically healed (the wound areas of S1 and S2 groups were 1 ± 0.5 mm^2^ and 1.8 ± 0.7 mm^2^, respectively), while the wound area of the control group was larger (14.3 ± 0.5 mm^2^). In addition, the pathological results after 20 d showed that there was inflammation in the control group, but there was no inflammation in S1 and S2 groups [66]. PCL/ type I collagen nanofiber membranes with different fiber spatial arrangements (random, aligned and crossed) were prepared by uniaxial electrospinning. The prepared fiber membranes were used to treat the wounds of diabetic rats. After 7 d of treatment, the wound healing rate of diabetic rats was 70% in the crossed group, 62% in the aligned group, and 56% in the random group, while it was only 40% in the control group. After 14 d, the wound healing rate of the crossed group was the highest, in excess of 95%. The results showed that the fiber arrangement has a great influence on the diabetes wound-healing effect [67].

Natural polymer gels can maintain the moist environment of a wound and promote wound healing, but Gel fiber immediately dissolves after contact with water and loses its fiber form [68]. To solve the problem of Gel nanofibers ready solubility in water, Sanhueza used a poly-3-hydroxybutyrate (PHB) (8% *w*/*v*) chloroform solution and Gel (30% *w*/*v*) acetic acid solution as spinning solutions, and dual-sized Gel/PHB nano/microfibers (Gel/PHB) was prepared by dual-jet electrospinning with double needles. The wounds of diabetic rats were treated with the Gel/PHB fiber membrane as well as the PHB micron fiber membrane and Gel nanofiber membrane obtained by uniaxial electrospinning, and the wounds treated only with saline solution were used as a control. The results showed that the Gel fibrous membrane immediately dissolved after contact with tissue fluid, and the formed viscous substance was left in the wound, which hindered the diabetes wound healing while there was no viscous substance formed in the wound treated with the Gel-PHB fibrous membrane. After 7 d, the wound healing rate of the Gel-PHB group was 30%, while that of the control group and the PHB group were 28 and 26% respectively [69]. 

SS is a kind of natural protein biomaterial, which has great potential in tissue regeneration due to its excellent antioxidant and antibacterial activity. A high level of ROS will delay the process of wound healing, and the antioxidant activity of SS helps eliminate ROS produced by senescent cells in the process of chronic inflammation, thus promoting wound healing. Gilotra prepared PVA/SS nanofiber films with fiber diameters of 130–160 nm by uniaxial electrospinning. This fibrous membrane can slowly release SS for 28 d, which can promote the transition of the wound healing process from the inflammation stage to the proliferation stage, which is conducive to diabetic wound healing [70]. Chouhan first mixed a PVA solution (13% *w*/*w*) and *Antheraea assama* silkworm silk fibroin (AaSF) solution (3% *w*/*w*) in equal volumes to obtain a PVA/AaSF solution, then an AaSF nanofiber membrane was prepared by uniaxial electrospinning, and then recombinant spider silk fusion proteins (FN-4RC and Lac-4RC) were coated on the AaSF fiber membrane to obtain the AaSF-FN-Lac film. The prepared fiber membranes ((1) AaSF membrane, (2) AaSF-FN membrane (coated only with FN-4RC), (3) AaSF-Lac membrane (coated only with Lac-4RC), and (4) AASF-FN-Lac membrane) were used to treat the wounds of diabetic rabbits. Commercially available wound dressing Duoderm and the untreated group (UNT) were used as controls. The AASF-FN, AASF-Lac, and AASF-FN-Lac groups promoted faster wound healing than the AaSF and Duoderm groups. Among all treatment groups, the ratio of remaining wound area after 12 d was 8% in the AASF-FN-Lac group, 15–18% in the AASF-FN and AASF-Lac groups, 24 ± 2.09%, 69 ± 6.45%, and 88 ± 6.39% in the AaSF, Duoderm, and UNT groups. Wound healing in the AASF-FN-Lac group was the fastest, with wounds completely healing within 14 d. The results showed that recombinant spider silk fusion proteins can promote granulation tissue formation, re-epithelialization, and ECM deposition at the wound, resulting in a better diabetic wound healing effect [71]. The experimental schematic diagram is shown in Figure 3.

### 3.2. Nanoparticle-Loaded Electrospun Fibers in the Treatment of Diabetic Wounds

The electrospun polymer fibers loaded with β-glucan (βG), copper-based metal-organic framework (MOFs), bioactive glass (BGs), sodium percarbonate (SPC), cerium oxide nanoparticles (nCeO_2_), and other nanoparticles can be used in the treatment of diabetic wounds (as shown in Table 1).

#### 3.2.1. Nanoparticles/Synthetic Polymer Electrospun Fibers

βG can activate the innate immune system by binding to Dectin-1 receptors on macrophages, dendritic cells, and neutrophils, which contributes to the transformation of M1 macrophages into M2 macrophages and promotes chronic wound healing. Grip added βG into the mixed solution of PEO and HPMC to prepare a spinning solution. HPMC/PEO nanofiber films loaded with βG were prepared by uniaxial electrospinning. The wound- healing effect of nanofibers was evaluated using the wounds of male diabetic mice. Diabetic mice were randomly divided into six experimental groups: four groups were treated with nanofibers, one group was injected with 50 μL water as the negative control group, and one group was injected with 50 μL growth factor solution (10 μg PDGF and 1 μg TGF-α dissolved in 0.5% (*w*/*v*) hydroxypropyl methylcellulose solution) as the positive control group. The therapeutic effects of three different doses of βG nanofibers (containing 190, 370, and 990 μg βG, respectively) and blank HPMC/PEO nanofibers without βG were evaluated. The results showed that the wound healing of the four nanofiber groups was better than that of the negative control group. After 4 d, the remaining wound area ratio of the βG nanofiber group was 76.8–82.3%, which was lower than that of the positive control group (97.9%), indicating that βG nanofibers can promote diabetic wound healing [72].

In the process of wound healing, the introduction of exogenous NO can promote angiogenesis and collagen deposition in the wound. However, high concentrations of NO (over 400 nM) can lead to apoptosis, which is not conducive to wound healing. In order to regulate the release behavior of NO, Zhang first prepared MOFs (HKUST-1) nanoparticles, and then the prepared HKUST-1 and 4-MAP were solvothermally reacted in a reactor to prepare secondary-amino-modified HKUST-1 nanoparticles. The modified HKUST-1 nanoparticles were activated at 120 °C for 10 h in a vacuum, cooled to room temperature, and then exposed to NO for 1 h at a pressure of 2 atm by a pressurizing device to load NO, and NO@HKUST-1 was obtained. NO@HKUST-1 was dispersed in HFIP, and then PCL was added and stirred to obtain the core layer spinning solution, while gel was dissolved in HFIP to obtain the shell layer spinning solution. NO@HKUST-1/PCL/Gel nanofiber films with a core/shell structure were prepared by coaxial electrospinning. The fiber membrane can release NO for 14 d with an average release rate of 1.74 nmol/l/h. The wounds of diabetic mice were treated with PCL/Gel fiber membranes (PG), HKUST-1/PCL/Gel (HPG), and NO@HKUST-1/PCL/Gel (NO@HPG) fiber membranes, respectively, untreated diabetic wounds were the control group. The results showed that the NO@HPG nanofiber membrane can promote angiogenesis and inhibit inflammation. NO and Cu^2+^ released by NO@HPG nanofiber membranes can cooperatively promote endothelial cell growth. The wound healing rates of the NO@HPG group at 11 and 13 d were 97.80 and 99.57%, respectively, which were significantly higher than those of other groups (HPG: 87%, 89%, PG: 77%, 81%, Control group: 62%, 81%) [73]. The preparation process of NO@HPG fiber membranes and their mechanism of promoting diabetic wound healing are shown in Figure 4.

BGs can change the cell microenvironment by releasing inorganic ions (such as Si^4+^), and Si^4+^ can stimulate expression of HIF-α, thereby promoting endothelial angiogenesis, which is conducive to diabetic wound healing. Elshazly prepared nanofibers loaded with BGs (BGnf) by uniaxial electrospinning. The prepared BGnf was applied to wounds of diabetic rabbits, and untreated wounds of diabetic rabbits were used as the control group. Immunohistochemical analysis showed that the percentage of VEGF expression in the BGnf group (14.08 ± 3.88%) was higher than that in the control group (3.92 ± 0.221%) after one week. After three weeks, the percentage of VEGF expression in the BGnf group (18.48 ± 1.458%) was increased compared to that in the control group (16.81 ± 1.65%). After three weeks, the wounds in the BGnf group were completely closed and new blood vessels were formed, and there was no inflammation. In the control group, there were purulent exudates and less neovascularization in the wounds [74]. Jiang prepared polydopamine (PDA)-modified PLA/PCL fiber loaded with BGs (BGs/PDA/PM) by uniaxial electrospinning combined with a PDA coating method. The cumulative release concentration of Si^4+^ of BGs/PDA/PM was 0.517 μg/mL, 1.347 μg/mL, and 2.416 μg/mL on d 1, 3 and 7, respectively. PLA/PCL fiber (PM) (uniaxial electrospinning PLA/PCL solution), PDA modified PLA/PCL fiber (PDA/PM), and BGs/PDA/PM fiber were used to treat the wounds of diabetic mice, and the wounds of untreated diabetic mice were the control. After 7 d, the wound healing rates of the control group and the PM group were 48.9 and 43.7% respectively, which was significantly higher than that of the PDA/PM group (34.7%) and the BGs/PDA/PM group (24.8%). After 15 d, the wounds in the BGs/PDA/PM group basically healed, and the residual wound area rate in the BGs/PDA/PM group was the lowest (0.98%), followed by the PDA/PM group (1.33%), the PM group (4.83%), and the control group (8.13%). The above results suggest that the release of Si^4+^ from BGs/PDA/PM fibers can effectively treat diabetic wounds [75]. The effects of PM, PDA/PM, and BGs/PDA/PM on wound tissues are shown in Figure 5.

Hypoxia is one of the main causes of poor vascularization in diabetic wounds. Hypoxia leads to the lack of HIF-1α, while HIF-1α can regulate oxygen homeostasis and a long-term hypoxic environment caused by impaired blood supply in diabetic wounds. The lack of HIF-1α will make chronic wounds difficult to heal. Zehra prepared PCL nanofiber membrane loaded with SPC (PCL-SPC) by uniaxial electrospinning, which can produce oxygen continuously for 10 d. Diabetic rats were randomly divided into three groups: the untreated diabetic wound control group, the PCL nanofiber treatment group (uniaxial electrospinning PCL solution), and the PCL-SPC fiber treatment group. A wound healing experiment showed that, compared with the control group and the PCL fiber group, the PCL-SPC fiber group can effectively improve the structure of the epidermis and dermis, and accelerate the rate of epithelialization and wound healing. In addition, the relative expression of the HIF-1α gene was analyzed by quantitative polymerase chain reaction. The results showed that expression of the HIF-1α gene in the PCL-SPC group was 2.52 ± 0.26 times higher than that in the control group and was significantly higher than that in the PCL fiber group (expression of the HIF-1α gene in the PCL fiber group was 1.68 ± 0.03 times higher than that in the control group). The oxygen supply of SPC plays an important role in diabetic wound healing [76].

CeO_2_ has antibacterial activity, and electrospun nanofiber membranes loaded with CeO_2_ can also promote diabetic wound healing [82]. Augustine used an ultrasonic device to disperse nCeO_2_ in a mixture of chloroform/dimethylformamide (v: v = 9:1) and dissolved PHBV in this solution to obtain 20% *w*/*v* PHBV electrospinning solution. PHBV fiber membranes loaded with nCeO_2_ were prepared by uniaxial electrospinning, which was used to treat diabetic wounds. The fiber membrane with a nCeO_2_ loading of 1% (weight ratio) was PHBV/nCeO_2_-1. The results showed that the wound healing rate of the PHBV/nCeO_2_-1 group was significantly higher than that of the PHBV fiber group without nCeO_2_. On d 10, 20, and 30, the wound healing rate of the PHBV/nCeO_2_-1 group was 52%, 73%, and 80%, respectively, while that of the PHBV fiber group was 27%, 43%, and 69%, respectively. PHBV fiber membranes loaded with nCeO_2_ can promote diabetic wound healing [77]. The healing of wounds treated with PHBV fiber membranes and PHBV/nCeO_2_-1 fiber membranes are shown in Figure 6.

#### 3.2.2. Nanoparticles/Synthetic Polymers/Natural Polymer Electrospun Fibers

Zhang used a PLA solution containing BGs as the electrospinning solution of core layer, and an HFIP solution of Gel as the electrospinning solution of shell layer. The core/shell structure patterned BG@PLA/Gel (BG@PG) electrospun nanofiber membrane was prepared by using a honeycomb structure receiver through coaxial electrospinning. The wounds of diabetic mice were treated with BG@PG fiber membranes, disordered PLA/Gel electrospun fiber membranes (UPG), and patterned PLA/Gel electrospun fiber membranes (PG), respectively, and untreated diabetic wounds were used as the control group. The wound area of the PG group and the BG@PG group at 7 d was 25.4 mm^2^ and 24.9 mm^2^, respectively, which were significantly smaller than that of the control group (33.7 mm^2^) and the UPG group (30.2 mm^2^). At 14 d, the wounds of the BG@PG group were almost completely healed, while the wound area of the control group, UPG group, and PG group was 6.7 mm^2^, 4.0 mm^2^, and 2.1 mm^2^, respectively [78]. Compared with the disordered nanofiber membrane, the patterned fiber membrane can provide more adhesion sites and growth space for cells, promote the adhesion and proliferation of cells on the fiber membrane, and stimulate the growth and differentiation of cells. A three-trilayer nanofibrous membrane (BGs-TFM) was prepared by the continuous uniaxial electrospinning method, with a CS fiber membrane as the lower layer, CS and PVA as the middle layer, and PVA/BGs as the upper layer. The BGs-TFM fiber membrane has good biocompatibility, strong antibacterial activity, and can promote skin regeneration. The wound model of diabetic mice showed that BGs-TFM can up-regulate growth factors, such as VEGF and TGF-β, and down-regulate inflammatory factors, such as TNF-α and IL-1β, promote epithelial regeneration and collagen deposition, and thus promote wound healing [79].

Using triethyl phosphate, tetraethyl orthosilicate, and calcium nitrate tetrahydrate as raw materials, Lv synthesized NAGEL with a particle size less than 2 mm using sol-gel method, and then prepared PCL/Gel nanofiber films loaded with NAGEL by uniaxial electrospinning. Wound experiments in diabetic mice showed that fibrous membranes can promote diabetic wound healing by promoting angiogenesis, collagen deposition and re-epithelialization, and inhibiting inflammatory reactions. The nanofibers containing 0 and 10% NAGEL particles were PL and 10NAG-PL, respectively. The wounds of diabetic mice were treated with PL and 10NAG-PL, and untreated diabetic wounds were the control group. At 7 d, the wound area of the 10NAG-PL group decreased by 57%, which was higher than that of the control group (18%) and the PL group (42%). After 13 d, the wound healing rate of the 10NAG-PL group was 94%, which was significantly higher than that of the PL group (82%) and the control group (69%) [80]. Ahmed prepared the CS/PVA/ZnO nanofiber membrane and the CS/PVA nano-fiber membrane by electrospinning CS/PVA solution with or without ZnO through uniaxial electrospinning. The wounds of diabetic rabbits were treated with CS/PVA and CS/PVA/ZnO nanofiber membranes. The results of wound healing experiments in diabetic rabbits showed that the wound healing rate of the CS/PVA/ZnO nanofiber membrane group was 44.8 ± 4.9% after 4 d treatment, which was higher than that of the CS/PVA nanofiber membrane group (22.5 ± 3.0%). In addition, the wound healing rate of the CS/PVA/ZnO nanofiber membrane group was 90.5 ± 1.7% on the 12th d, which was much higher than that of the untreated diabetic wound control group (52.3 ± 2.8%) [81].

### 3.3. Drug-Loaded Electrospun Fibers for Collaborative Therapy of Diabetic Wounds

Drugs that promote diabetic wound healing including natural medicines (e.g., sesamol, epigallocatechin-3-gallate (EGCG)), platelet lysate (PL), growth factors, antimicrobial peptide, antibiotics (e.g., ciprofloxacin (CFX), doxycycline (DCH), tetracycline, vancomycin, gentamicin), small molecular inhibitors, non-sulfonylurea drugs (e.g., repaglinide), cytokines, protein agonists, thiazolidinediones (e.g., pioglitazone) and polymers, can be combined to prepare drug-loaded electrospinning fibers, which can not only have therapeutic effects of drugs, but also take advantage of the structural advantages of fibers to promote the healing of diabetic wounds (as shown in Table 2).

#### 3.3.1. Drugs/Natural Polymer Electrospun Fibers

Liu prepared CA/zein nanofiber membranes loaded with different masses of sesamol through uniaxial electrospinning and studied the effect of CA/zein fiber membranes on wound healing in diabetic mice. Diabetic mice were divided into five groups: Group C (normal mouse wound, control group), Group S (diabetic mouse wound, untreated), Group M (diabetic mouse wound, treated with blank nanofiber membranes without sesamol), Group L (diabetic mouse wound, treated with CA/zein nanofiber membranes loaded with 2% sesamol), and Group H (diabetic mouse wound, treated with CA/zein nanofiber membranes loaded with 5% sesamol). After 5 d, the wound healing rates of each group (C, S, M, L, and H) were 80%, 20%, 40%, 60%, and 70%, respectively. After 9 d, the wound healing rates of each group (C, S, M, L, and H) were 100%, 60%, 85%, 95%, and 100%, respectively. The wounds in groups L and H were similar to those in group C, and essentially had healed on d 9. Studies have shown that sesamol can down-regulate expression of inflammatory factors such as IL-1β, and TNF-α, while up-regulating expression of IL-6 (anti-inflammatory factor), which can promote the rapid healing of diabetic wounds [83].

#### 3.3.2. Drugs/Synthetic Polymer Electrospun Fibers

PL can release PDGF and VEGF, thus promoting collagen deposition and re-epithelialization to promote diabetic wound healing, but PL is easily inactivated when directly used. In order to improve the stability of PL, Losi prepared a protein/poly(ether)urethane fiber loaded with PL (FB-PL fiber) using uniaxial electrospinning. The wounds of diabetic mice were treated with FB-PL fibers, and the wounds treated with mepore polyurethane film (transparent breathable dressing) were used as the control group. After 14 d, the remaining wound area in the FB-PL group was 20%, which was much lower than that in the control group (78%). The cumulative release of PDGF and VEGF from FB-PL fiber membranes was detected by the ELISA method. The results showed that 40% growth factor was released from FB-PL fiber membranes on the first d and 80% on the 7th d [84]. CTGF is unstable in a highly oxidized diabetic wound environment, and the use of nanofiber membranes loaded with CTGF can improve its stability and promote diabetic wound healing [99]. Augustine mixed PVA aqueous solution with CTGF solution to prepare PVA solution (6%, *w*/*v*) containing CTGF (0.1wt%). At the same time, PLA in dichloromethane (DCM)/DMF (*v*/*v* = 1:9) solution was prepared. The core/shell PVA-CTGF/PLA nanofiber membrane was constructed by coaxial electrospinning using PVA solution with CTGF as the core layer spinning solution and PLA solution in DCM/DMF as the shell spinning solution. PVA-CTGF/PLA nanofiber membranes can slowly release CTGF for 15 d. In vitro wound healing experiments showed that the fibroblast wound shrinkage rate of the control group (the untreated diabetic wound group) was 32.51 ± 6.44%, while that of the PVA-CTGF/PLA group was 54.34 ± 6.8%. The keratinocyte wound shrinkage rate in the control group was 8.62 ± 2.34%, while that in the PVA-CTGF/PLA group was 45.54 ± 6.68%. The wound shrinkage rate of endothelial cells in the control group was 43.45 ± 4.58%, while that in the PVA-CTGF/PLA group was 58.64% ± 3.46%. Cell activity test results showed that compared with the control group, the number of living cells (fibroblast, keratinocytes, and endothelial cells) in the PVA-CTGF/PLA group was more, indicating that PLA/PVA-CTGF membrane is conducive to cell proliferation and migration, and beneficial to the treatment of diabetic wounds [85].

Su prepared PCL solution by dissolving PCL in a mixed solvent of DCM/DMF (*v*/*v* = 4:1), and an aqueous solution of antibacterial peptide 17BIPHE2 was added into the PCL solution to obtain electrospinning solution A, and an aqueous solution of Pluronic F127 was used as electrospinning solution B. 17BIPHE2-PCL/Pluronic F127 core/shell nanofiber membranes with 17BIPHE2-PCL as core layer and Pluronic F127 as shell layer were prepared by coaxial electrospinning of solutions A and B. Then 17BIPHE2 was coated on the surface of the 17BIPHE2-PCL/Pluronic F127 nanofiber membranes to obtain 17BIPHE2-PCL/F127-S fiber membranes, and 17BIPHE2-PCL/F127-S can continuously release antimicrobial peptide 17BIPHE2 for 28 d. PCL/F127 core/shell nanofiber films were prepared by coaxial electrospinning using PCL solution without antibacterial peptide 17BIPHE2 as the core layer spinning solution and B as the shell layer spinning solution. The wounds of diabetic mice were inoculated with 10 μL methicillin-resistant *Staphylococcus aureus* (MRSA) at a concentration of 1 × 10^8^ CFU/mL, and then the wounds were treated with PCL/F127 and 17BIPHE2-PCL/F127-S fiber membranes. In vivo antibiofilm efficacy test results showed that without debridement, 6.17 × 10^6^ CFU/g MRSA was detected at the wound site after 3 d of 17BIPHE2-PCL/F127-S fiber membrane treatment, which had 3.08 log reduction compared to the PCL/F127 control group. After debridement, no colony was found in wounds treated with 17BIPHE2-PCL/F127-S fiber membranes after 3 d, which was 9.86 log reduction compared to the PCL/F127 control group. These results indicate that bacterial biofilms in diabetic wounds can be eliminated after 3 d of debridement and 17BIPHE2-PCL/F127-S treatment, thus promoting diabetic wound healing [86]. Mabrouk first prepared PAA ethanol solution (7% *w*/*v*), PVP ethanol solution containing CFX (PVP content 20% *w*/*v*, CFX/PVP, m/m = 1:10, 1:20 and 1:30), and PCL ethanol solution (10% *w*/*v*). A three-layer nanofiber membrane (PAA/PVP/PCL nanofiber membrane) was prepared by continuous uniaxial electrospinning with a PAA fiber membrane as the lower layer, PVP/CFX as the intermediate layer, and PCL as the upper layer. The fiber membrane can continuously release CFX for 48 h and has antibacterial activity against gram-negative bacteria and gram-positive bacteria. The antibacterial properties of the PAA/PVP/PCL nanofiber membrane give it the potential to promote diabetic wound healing [87].

DCH is a broad-spectrum antibiotic and MMPs inhibitor. Local administration of DCH can be used to treat chronic wounds, but local administration of DCH has some problems, such as poor efficacy and strong skin irritation. Cui added DCH into a PLA/HFIP solution, stirred at room temperature for 30 min to prepare the DCH contained polymer solution, and then prepared DCH-loaded PLA nanofiber membranes (DCH/PLA) by uniaxial electrospinning. The fiber membrane can release DCH for two weeks. The wounds of diabetic rats were treated with normal saline (the control group), uniaxial electrospun PLA nanofiber membranes (the PLA group), DCH solution dropping combined PLA nanofiber membranes (the DCH+PLA group), and DCH/PLA nanofiber membranes (the DCH/PLA group). After 7 d, the wound area of the PLA group (44.3 ± 5.9 mm^2^) was almost the same as that of the control group (47.4 ± 2.6mm^2^). In the DCH+PLA group, the wound was significantly reduced when the DCH concentration was 10 and 15% (the wound area was 16.6 ± 3.6 mm^2^ and 18.1 ± 4.4 mm^2^, respectively), but when the DCH concentration was increased to 20%, the wound healing speed significantly decreased (the wound area was 29.3 ± 9.6 mm^2^). The wound area of the DCH/PLA group on the 7th d was only 6.3 ± 2.7 mm^2^, indicating that loading DCH into PLA fibers can effectively improve the therapeutic effect of DCH on diabetic wounds [88]. Alhusein dissolved PCL in CHCl_3_/MeOH (*v*/*v* = 9:1) to prepare PCL solution and added MeOH solution of tetracycline (Tet) into the above PCL solution to prepare the PCL electrospinning solution containing 3% *w*/*w* Tet (solution A). Then polyethylene-co-vinyl acetate (PEVA) was dissolved in CHCl_3_/MeOH (*v*/*v* = 9:1) to obtain the PEVA solution. The MeOH solution of Tet was added to the PEVA solution to obtain the PEVA electrospinning solution containing 3% *w*/*w* Tet (solution B). The three-layer PCL/PEVA/PCL fiber membrane was prepared by uniaxial electrospun solution A and B successively. This fibrous membrane can continuously release Tet for 14 d, which can effectively inhibit the formation of *Staphylococcus aureus* bacterial membranes and kill bacteria, thus promoting diabetic wound healing [100].

DMOG is a non-specific small-molecule inhibitor of prolyl hydroxylases, which can inhibit the decomposition of HIF-α to create a cell microenvironment similar to hypoxia. In this micro-hypoxia environment, angiogenesis and fiber regeneration are activated, thus accelerating the rate of wound healing [101]. Zhang prepared DMOG-loaded PCL fiber membranes (PCLF/DMOG) and drug-free PCL fiber membranes (PCLF) by electrospinning. The wounds of diabetic rats were treated with PCLF membranes and PCLF/DMOG membranes, and untreated diabetic wounds were the control group. The wound healing rates of the control group on the 3rd, 9th, and 14th d were 7%, 56%, 70%, and 11%, 60%, 75% in the PCLF group, while those in the PCLF/DMOG group were 20%, 62%, and 89%, respectively. Histological analysis showed that the rate of re-epithelialization after 14 d in the control group was 47%, while that in the PCLF group was 50% and that in PCLF/DMOG group was 75%. The high rate of re-epithelialization and wound healing of the PCLF/DMOG group indicated that PCLF/DMOG can promote diabetic wound healing [89]. Ren first prepared DMOG-loaded mesoporous silica nanospheres (DS), then prepared PLLA nanofiber membrane loaded with DS (10 DS-PL) through uniaxial electrospinning, and the PLLA electrospinning membrane (PL) without drug loading was used as the control membrane. Wounds of diabetic mice were treated with PL and 10 DS-PL. After 11 d, the wound healing rates of the PL group and the 10 DS-PL group were 76 and 82%, respectively, while those of the untreated diabetic wound group were only 70%. After 15 d, the wound healing rate in the 10 DS-PL group was 97%, which was higher than that in the PL group (94%) and the untreated diabetic wound group (84%) [102].

In order to solve the problems of poor water solubility and unstable drug absorption of repaglinide, Thakkar prepared drug-loaded PVA/PVP nanofibers by uniaxial electrospinning of the PVA/PVP electrospinning solution containing repaglinide and investigated their effect on diabetic wound healing. Four groups of experiments were designed and casted film was prepared with the same polymer solution. Diabetic rats were randomly divided into four groups: the first group was untreated diabetic wounds (the control group), the second group was the repaglinide group, the third group and the fourth group were PVA/PVP electrospinning nanofiber group loaded with repaglinide and the casted film group loaded with repaglinide (obtained by the film casting method). The drug release experiment showed that the drug release amount of the third group and the fourth group after 10 min was 90 and 73%, respectively, while that of the second group was only 10%. The oral glucose tolerance test showed that the glucose level of nanofibers was (107.66 ± 6.72 mmol/L) after 120 min, which was lower than that of the control group (154.66 ± 6.47 mmol/L), the repaglinide group (142.33 ± 5.817 mmol/L), and the casted film group (110.00 ± 15.55 mmol/L). The above experimental results show that repaglinide-loaded nanofibers release more repaglinide, and the blood glucose level decreases, which is beneficial to diabetic wound healing [90].

Drug combination therapy can more effectively promote diabetic wound healing. Lee dissolved PLGA, vancomycin, and gentamicin in HFIP to produce the PLGA-antibiotic solution as the shell spinning solution, and dissolved PDGF in PBS to produce the core spinning solution. Core/shell structure PDGF/PLGA-antibiotic nanofibers (A) were prepared by coaxial electrospinning. PBS/PLGA/antibiotic nanofibers (B) with a core/shell structure were prepared by coaxial electrospinning with PBS solution as the core spinning solution and PLGA/antibiotic as the shell spinning solution. Antibiotic/PLGA nanofibers (C) were prepared by uniaxial electrospinning of the PLGA-antibiotic solution. A nanofiber can continuously release antibiotics (e.g., vancomycin, gentamicin) and PDGF for more than 3 weeks. The wound healing experiment in diabetic mice showed that the wound area of fiber A (20.4 ± 1.7 mm^2^) was significantly smaller than that of fiber B (26.4 ± 1.0 mm^2^) and fiber C (26.4 ± 1.0 mm^2^) after 7 d. After 14 d, the wound area in fiber B and C was reduced to 14.0 ± 0.7 mm^2^ and 20.8 ± 1.3 mm^2^, respectively, while the wound area in fiber A was only 12.2 ± 0.1 mm^2^. PDGF/PLGA/antibiotic nanofiber membrane can promote angiogenesis and epidermal hyperplasia through the synergistic effect of PDGF and antibiotics to improve the diabetes wound healing effect [91]. The preparation of PDGF/PLGA/antibiotic nanofibers and the mechanism of promoting diabetic wound healing are shown in Figure 7.

Dwivedi prepared, respectively, Eudragit RL/RS nanofiber membrane loaded with GS (A) and Eudragit RL/RS nanofiber membrane without GS (B) by uniaxial electrospinning. Then rhEGF was immobilized on the surface of membrane A by the covalent immobilization technique to obtain GS and rhEGF co-loaded Eudragit RL/RS nanofiber membrane (C). In the wound healing experiment of diabetic mice, five groups were designed, the first group was untreated diabetic wounds (the negative control group), the second group was wounds treated with GS solution (the positive control group), the third group was wounds treated with B fiber membranes, the fourth group was wounds treated with C fiber membranes, and the fifth group was wounds treated with A fiber membranes. The experimental results showed that the residual wound area rates in the fourth group at 4, 8, and 12 d were 14.31 ± 2.61%, 10.76 ± 1.92%, and 8.91 ± 1.95%, respectively, which were much lower than those in other groups (the first group: 94%, 92%, and 89%, the second group: 55%, 50%, and 48%, the third group: 91%, 90%, and 88%, the fifth group: 64%, 60%, and 58%), indicating that GS and rhEGF have a better synergistic effect in the treatment of diabetic wounds [92].

#### 3.3.3. Drugs/Synthetic Polymer/Natural Polymer Electrospun Fibers

Chemotactic cytokines are a kind of small molecular cytokine that can cause chemotactic responses. MCP-1 is a chemotactic cytokine that can promote macrophages to participate in the process of wound healing. Yin prepared respectively MCP-1-loaded Gel-PGA nanofiber membrane (DES) and cytokine-free Gel-PGA nanofiber membrane (NES) by uniaxial electrospinning, DES and NES were used to treat the wounds of diabetic mice. After 3 d, the number of F4/80+ macrophages in the DES group were 1400 cells/mm^2^, which was much higher than that of 750 cells/mm^2^ in the NES group and 800 cells/mm^2^ of the control group. After 5 d, the wound closure rate of the DES group was 48.34 ± 10.23%, the NES group was 73.27% ± 11.45%, and the untreated diabetic wound group was 80.27 ± 15.56%. The wounds of the DES group fully recovered after 10 d, while the wounds of the untreated diabetic wound group needed 14 d to fully recover [93].

EACCs can promote diabetic wound healing, but the survival rate of EACCs is low under a high glucose environment. SRT1720 can improve the survival rate of EACCs, and then promote the healing of diabetic wounds [103,104]. Cheng prepared PLGA-collagen protein-silk nanofiber membranes loaded with SRT1720 (PCSS) by uniaxial electrostatic spinning. EACCs (5 × 10^5^) were inoculated on the surface of PCSS fiber membranes to obtain PCSS-EACCs. PCSS-EACCs can steadily release SRT1720 at a rate of about 7.14% per d for 15 d, thus promoting the release of VEGFA and IL-8 from EACCs. The released VEGFA and IL-8 can promote endothelial cell proliferation, migration, and angiogenesis. The wounds of diabetic mice were treated with PCSS-EACCs, and the wounds of normal mice were the control group. After 14 d, the wound healing rate of the PCSS-EACCs group was fast, which was similar to that of normal mice, and the wound residual area rate of the PCSS-EACCs group was 2% [94].

Pioglitazone is a thiazolidinedione antidiabetic drug, which is an insulin sensitizer. Pioglitazone can activate the peroxidase-activated receptor PPAR-γ, thereby regulating the transcription of insulin-related genes that control glucose and lipid metabolism and maintaining normal blood glucose level to promote diabetic wound healing. Yu first prepared a formic acid/acetic acid solution of PCL as spinning solution A, and then dissolved Gel and pioglitazone in the mixed solvent of formic acid/acetic acid (*v*/*v* = 7:3) and stirred for 2 h to prepare spinning solution B. Using nylon mesh with a fixed pore size of 40 μm as the receiving device, the micropatterned PCL nanofiber membrane was prepared by electrospinning solution A, and then electrospinning solution B, and the fiber was further deposited on the PCL film to prepare PCL/Gel-pioglitazone nanofiber membranes (PCL/Gel-pio). PCL/Gel-pio has an asymmetric hydrophobic outer layer and a hydrophilic inner layer, which can effectively simulate the epidermis and dermis of natural skin (Figure 8A). The prepared PCL/Gel-pio fiber membrane was stripped from nylon mesh and soaked in ethanol solution containing 2% *w*/*w* genipin to undergo the genipin-based cross-linking reaction, which can improve the stability of the PCL/Gel-pio fiber membrane (Figure 8B). The cross-linked PCL/Gel-pio nanofiber membrane can promote diabetic wound healing by preventing bacterial adhesion and controlling the release of pioglitazone (Figure 8C). After the wounds of diabetic mice were treated with cross-linked PCL/Gel-pio nanofiber membranes, it was found that the fiber membranes significantly up-regulated expression of MIP-2, TNF-α, and VEGF in the wound on the 7th d, which could promote wound healing. After 10 d, it significantly reduced expression of MMP-9, IL-1β, and IL-6 at the wound site to reduce the inflammation of the wound. This fiber membrane can effectively promote wound healing in type 1 and type 2 diabetic mice [95].

Shin dissolved synthetic polymer PLGA and the natural drug EGCG in HFIP to prepare the shell spinning solution. The natural polymer HA aqueous solution was used as the core spinning solution. HA/PLGA core/shell nanofiber membranes loaded with EGCG (HA/PLGA-E) were prepared by coaxial electrospinning. The wounds of diabetic rats were treated with HA/PLGA-E nanofiber membranes, PLGA nanofiber membranes (prepared by uniaxial electrospinning PLGA solution), and HA/PLGA core/shell nanofiber membranes (prepared by coaxial electrospinning PLGA solution and HA aqueous solution). After two weeks, the residual wound area rate in the HA/PLGA-E group was 10.84%, which was significantly lower than that in the other groups (the untreated diabetic wound group, the PLGA group, and the HA/PLGA group were 49.96%, 48.43% and 40.18%, respectively) [96]. Cod liver oil can promote wound healing by increasing the blood supply of the wound and changing the phospholipid composition of the membrane [105,106]. Khazaeli first added a water/ethanol (*v*/*v* = 18:3) solution of PLA into a water/dimethylformamide (*v*/*v* = 2:1) solution of CS to prepare the polymer solution of PLA/CS. Then the PLA/CS electrospinning solution of 30%w/w cod liver oil was prepared by adding tween and cod liver oil into the above PLA/CS polymer solution and reflowing. The cod liver oil loaded PLA/CS nanofiber membrane was prepared by uniaxial electrospinning, which was used to treat diabetic mice wounds. Wound experimental results of diabetic mice showed that after 14 d, the wound healing rate of the fiber group was 94.5%, which was much higher than that of the free cod liver oil group (40%) and the untreated diabetic wound group (13%) [97].

Chouhan prepared PVA-SF nanofiber membranes co-loaded with EGF, bFGF and LL-37 antimicrobial peptide by uniaxial electrospinning. Different kinds of SF (*Bombyx mori* silk fibroin (BMSF), *A. assama* silk fibroin (AASF), and *P. ricini* silk fibroin (PRSF)) showed different effects on wound healing in diabetic rabbits. The wound healing rate of the AASF group and the PRSF group was 85–90% on the 14th d, which was significantly higher than that of the BMSF group (73%). LL-37 antimicrobial peptide can reduce inflammation in wounds, and EGF and bFGF can promote the proliferation of fibroblasts, keratinocytes, and endothelial cells, thus promoting the wound healing of diabetes. Histological analysis showed that granulation tissue regeneration, angiogenesis and re-epithelialization were faster in the AASF group and the PRSF group, which had a better effect on diabetic wound healing [98].

### 3.4. Drug/Nanoparticle/Polymer Electrospun Fibers

Studies have shown that polymer electrospun fibers loaded with antibacterial agents and zinc oxide nanoparticles (n-ZnO) have a positive effect on diabetic wound healing. Jafari added amoxicillin (AMX) (15wt%) into Gel solution and stirred for 1 h, then it was mixed with PCL solution to prepare electrospinning solution A, while n-ZnO (4wt%) in PCL solution was prepared for elctrospinning solution B. AMX-loaded PCL-Gel nanofiber film was prepared by uniaxial electrospinning of spinning solution A, and the fiber obtained by uniaxial electrospinning solution B was deposited on the AMX-loaded PCL-Gel membrane, thus, the n-ZnO-AMX double-layer nanofiber membrane was prepared. The drug release test in vitro showed that n-ZnO-AMX fiber membrane can slowly release AMX for 144 h. The antibacterial effect of AMX can reduce the inflammatory reaction of diabetic wounds and promote the transition of wound healing from the inflammation stage to the proliferation stage. The release of ZnO from n-ZnO-AMX fibrous membranes acts on the wounds to produce ROS, ROS initiates chemical reactions to promote the production of vascular regulatory growth factors, and, thus promotes angiogenesis. After 3 d, the wound healing rate of rats treated with n-ZnO-AMX fiber membranes (46.58 ± 3.66%) was significantly higher than that of the untreated diabetic wound control group (36.73 ± 4.93%). Histological analysis showed that n-ZnO-AMX fiber membranes can increase collagen deposition, promote neovascularization, and reduce scar formation under the synergistic effect of AMX and n-ZnO, thus promoting diabetic wound healing [107].

### 3.5. Cell Loaded Electrospun Fiber Membranes for Diabetic Wound Treatment

Cells can be cultured and induce differentiation on the electrospun fiber membrane, which can promote the healing of diabetic wounds by promoting angiogenesis of differentiated cells.

Bone marrow mesenchymal stem cells (BMSCs) can promote angiogenesis and thus promote diabetic wound healing, but BMSCs cannot survive in a high glucose environment, while Klotho protein has a protective effect on BMSCs under high glucose conditions [108]. Liu first prepared Klotho-protein-loaded CS microspheres, and then added the CS microspheres into an aqueous solution of Gel and stirred for 30 min to obtain the Gel solution containing CS microspheres. Then the solution was uniformly coated on the pre-prepared PLGA fiber membrane (the spinning solution was prepared by dissolving PLGA in the mixed solvent of CHCl_3_/DMF (*v*/*v* = 9:1), and the PLGA fiber was prepared by uniaxial electrospinning) with the coating method. After natural solidification of the Gel, the PLGA/Gel fibers were obtained. Then with the uniaxial electrospinning PLGA spinning solution, a PLGA layer was electrospun on PLGA/Gel fibers to obtain PLGA/Gel/PLGA nanofiber membranes (the structure of PLGA/Gel/PLGA fiber membrane is similar to the structure of a sandwich, that is, two layers of PLGA nanofiber membrane sandwich the intermediate Gel layer, and the Gel layer contains CS microspheres). Finally, BMSCs were inoculated on the surface of PLGA/Gel/PLGA nanofiber membranes to obtain Klotho + BMSCs nanofiber membranes. The Klotho + BMSCs fiber membrane can slowly release Klotho protein for 7 d. The results of EDU(5-ethynyl-2′-deoxyuridine) experiments showed that the proliferation rate of BMSCs was increased by 126% when Klotho protein was directly applied to diabetic wounds, indicating that Klotho protein can promote the proliferation of BMSCs under high glucose condition. The wounds of diabetic mice were treated with Klotho + BMSCs nanofiber membranes, Klotho, and BMSCs, respectively, and untreated diabetic wounds were used as controls. After 10 d, the wound healing rate in the BMSCs + Klotho group was 80%, which was higher than that in the Klotho group (16%) and the BMSCs group (17%), and much higher than that in the control group (39%). The results showed that when BMSCs cells were incubated on electrospun fibers, and the Klotho-protein-loaded electrospun fibers can promote the differentiation of BMSC cells, thus promoting angiogenesis in diabetic wounds, which can achieve effective diabetes wound healing [109].

## 4. Outlook

Compared with normal wounds, the healing process of diabetic wounds is often in the inflammatory stage for a long time due to uncontrolled inflammatory reaction, and the angiogenesis in the wound is difficult. In addition, there is a long-term and recurrent bacterial infection problem in diabetic wounds. Electrospun nanofiber membrane has great potential for application in the treatment of diabetic wounds because of its advantages in properties and structure.

The structure of electrospinning nanofibers is similar to the structure of ECM, which is conducive to the attachment, growth and migration of fibroblasts, thereby facilitating the formation of new skin tissue in the wound. The easily modified characteristics of electrospinning nanofibers favor their structural modification. The two-dimensional fiber membrane could be transformed into three-dimensional structure through multi-layer stacking, gas-foaming technique and other new methods, which is beneficial to improving the proliferation rate of cells, thus accelerating diabetic wound healing [110,111].

Electrospinning fiber has the advantage of easy loading. Recent studies have shown that M2 macrophages play an important role in diabetic wound healing. Electrospinning nanofiber membranes can directly promote the transformation of M1 macro-phages (pro-inflammatory macrophages) into M2 macrophages (anti-inflammatory macrophages), and then promote diabetic wound healing. Incubating cells on the electrospinning fiber membrane for diabetic wound healing is a hot research topic currently. It is expected that M2 type macrophages could be used for diabetic wound healing after incubating on the fiber membrane. In addition, silver nanoparticles have good antibacterial effects and can be used in the treatment of normal wounds; however there have been no reports about their treatment of diabetic wounds [112,113,114]. Based on the complex microenvironment of diabetic wounds, the preparation of electrospun fiber membrane, by combining silver nanoparticles with other active components, may be an effective method for the treatment of diabetic wounds.

Hydrogel fiber dressing is a new type of wound dressing, which has the advantages of high specific surface area, high liquid absorption, and good air permeability. Hydrogel fiber has the functional properties of hydrogels (e.g., high water content, high elasticity, and stimulus-response) and the structural advantages of fibers (e.g., high specific surface area and easy weaving). The development of electrospinning preparation methods for hydrogel fibers, and the combination of hydrogel fibers with active ingredients for the healing of diabetic wounds also provide an opportunity for the treatment of diabetic wounds.

It should be noted that there are many in vivo animal experiments performed in diabetic wound healing studies, and most of the healing effect is evaluated by healing time, while analysis of the diabetic wound healing process and tissue section is insufficient. The relative lack of research on the biocompatibility and biodegradability of electrospun fibers also creates challenges for their safety evaluation.

## Figures and Tables

**Figure 1 pharmaceutics-14-00006-f001:**
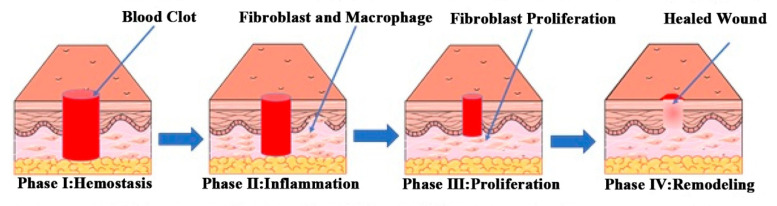
Schematic diagram of the normal wound healing process. Reproduced with permission from [17], ACS, 2019.

**Figure 2 pharmaceutics-14-00006-f002:**
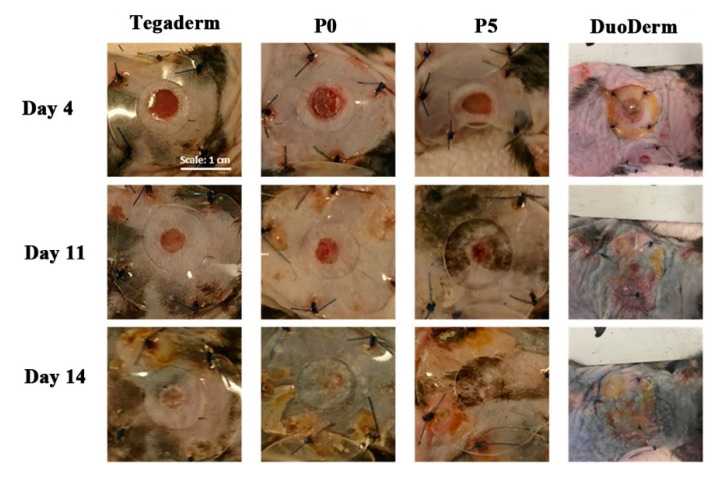
Photos of diabetic wounds treated with P5 fiber membranes at 4, 11, and 14 d. Reproduced with permission from [65], ACS, 2021.

**Figure 3 pharmaceutics-14-00006-f003:**
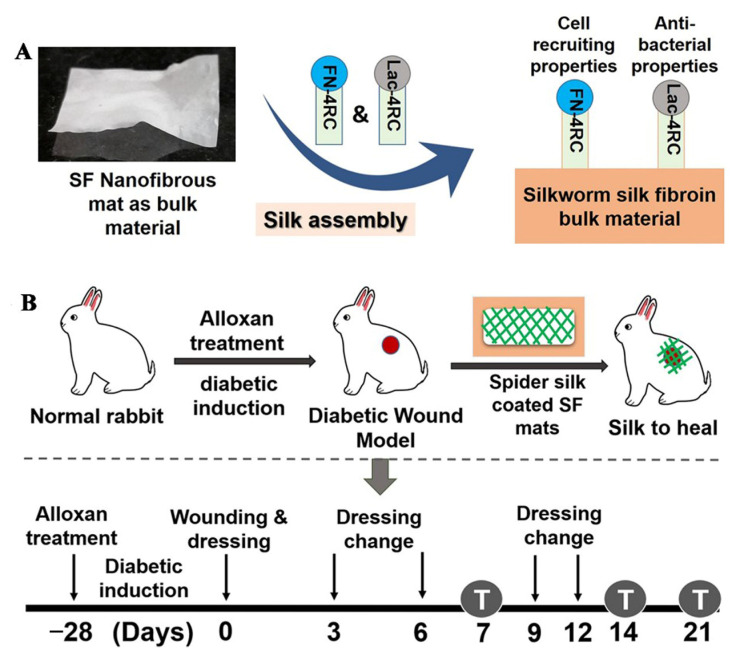
Schematic diagram of the experiment. (**A**) preparation of the nanofiber membrane, (**B**) procedure of wound treatment in diabetic rabbits. Reproduced with permission from [71], ACS, 2019.

**Figure 4 pharmaceutics-14-00006-f004:**
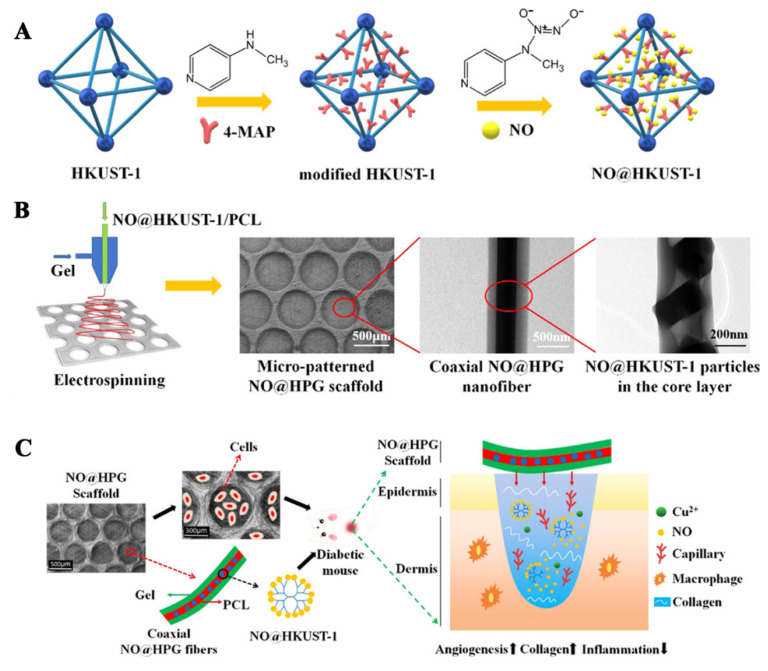
(**A**) Preparation process of NO@HKUST-1, (**B**) preparation and characterization of NO@HPG fiber membranes, (**C**) the diabetic wound healing mechanism of NO@HPG fiber membranes. Reproduced with permission from [73], ACS, 2020.

**Figure 5 pharmaceutics-14-00006-f005:**
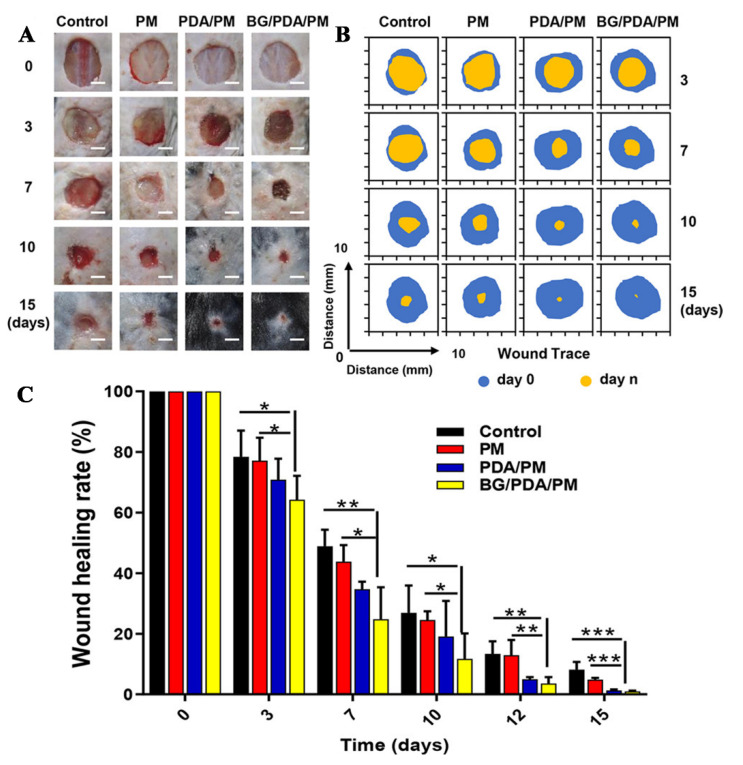
(**A**) Diabetic mice wound photos, (**B**) Trace of the wound area, (**C**) Wound healing rate after treatment in Ctrl, PM, PDA/PM, and BGs/PDA/PM groups at 0, 3, 7, 10, and 15 d. Data were presented as means ± standard error. Differences were considered significant when *p* < 0.05 (*), *p* < 0.01 (**), or *p* < 0.001 (***). Reproduced with permission from [75], ACS, 2020.

**Figure 6 pharmaceutics-14-00006-f006:**
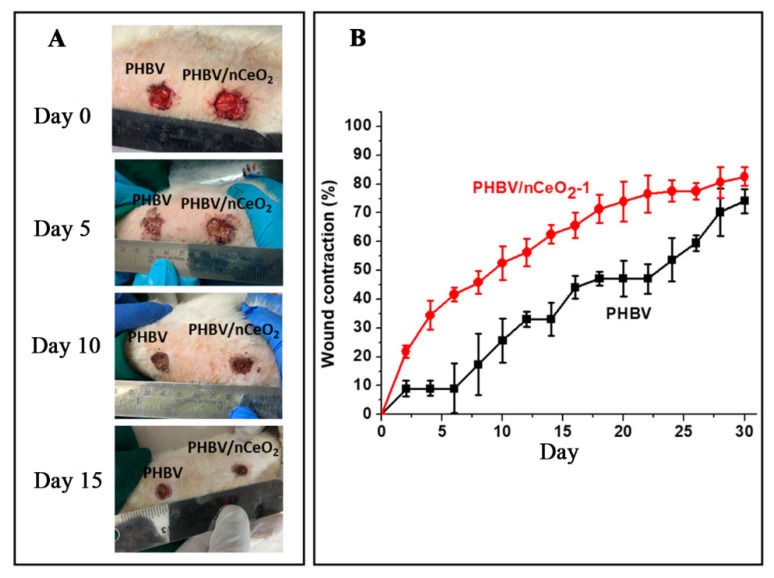
(**A**) Diabetic wound images, and (**B**) wound shrinkage curves treated with PHBV and PHBV/nCeO_2_-1 fibrous membranes. Reproduced with permission from [77], ACS, 2020.

**Figure 7 pharmaceutics-14-00006-f007:**
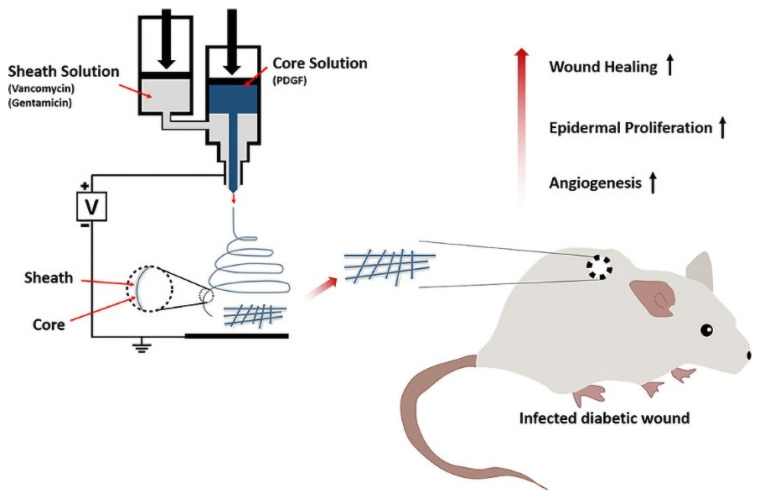
Preparation of PDGF/PLGA/antibiotic nanofibers and the mechanism of promoting diabetic wound healing. Reproduced with permission from [91], ACS, 2020.

**Figure 8 pharmaceutics-14-00006-f008:**
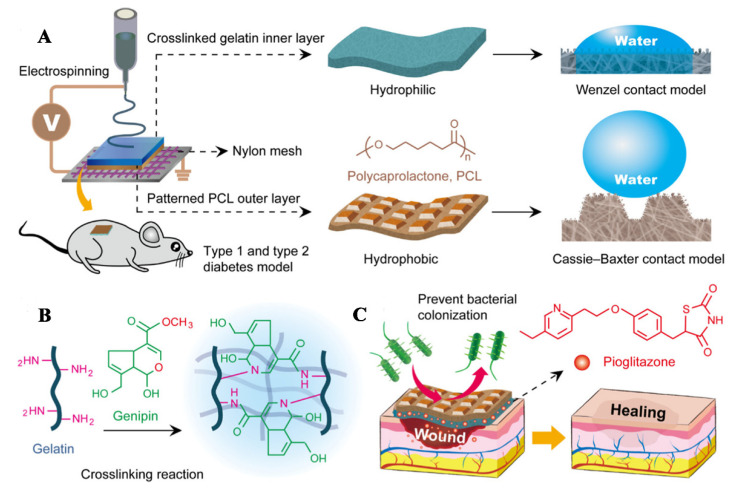
(**A**) Schematic diagram of the preparation of PCL/Gel-Pio nanofiber membranes, (**B**) The mechanism of the genipin cross-linking reaction of fiber membrane, and (**C**) The principle of fiber membrane promoting diabetic wound healing. Reproduced with permission from [95], ACS, 2020.

**Table 1 pharmaceutics-14-00006-t001:** Nanoparticle-loaded electrospun fibers in the treatment of diabetic wounds.

Name of Electrospun Fiber Membrane	Spinning Polymers	Active Ingredient	Mechanism of Action	Reference
βG-loaded hydroxypropyl methylcellulose (HPMC)/PEO nanofiber	HPMC, PEO	βG	βG activates the innate immune system by binding to dectin-1 receptors on macrophages, dendritic cells, and neutrophils to transform macrophages from M1 to M2.	[72]
NO@HKUST-1 (MOFs)/PCL/Gel nanofibrous membranes	PCL, Gel	NO-loaded HKUST-1 particles	NO@HKUST-1/PCL/Gel nanofiber membrane can promote angiogenesis and inhibit inflammation. Cu^2+^ released by HKUST-1 and NO can cooperatively promote endothelial cell growth.	[73]
BGs nanofibers (BGnf)	Polyvinyl butyral (PVB)	BGs	BGs can change the cell microenvironment by releasing Si^4+^, which stimulates expression of hypoxia inducible factor-α (HIF-α) and thus promotes the angiogenesis of endothelial cells	[74]
BGs loaded polydopamine (PDA)-modified polylactic acid (PLA)/PCL nanofibrous membranes (BGs/PDA/PM)	PLA, PCL	BGs	Si^4+^ released from BGs/PDA/PM nanofiber membranes can stimulate expression of HIF-α and promote angiogenesis	[75]
SPC-loaded PCL nanofibrous membranes	PCL	SPC	Hypoxia inducible factor-1α (HIF-1α) promotes diabetic wound healing by promoting angiogenesis. Long-term hypoxia will cause HIF-1α deficiency. The oxygen supply of SPC plays an important role in diabetic wound healing	[76]
nCeO_2_-incorporated poly (3-hydroxybutyrae-co-3- hydroxyvalerate) (PHBV) membranes	PHBV	nCeO_2_	During the inflammation phase, ROS produced by nCeO_2_ can inhibit bacterial growth and promote diabetic wound healing	[77]
BGs@PLA/Gel nanofibrous membranes	PLA, Gel	BGs	Si^4+^ released from BGs@PLA /Gel nanofiber membrane can up-regulate expression of hypoxia inducible factor-1 (HIF-1), and thus up-regulate expression of pro-angiogenic factors such as bFGF and VEGF	[78]
BGs-incorporated CS-PVA trilayer nanofibrous membrane (BGs-TFM)	PVA, CS	BGs	BGs-TFM up-regulates growth factors VEGF and TGF-β, down-regulates inflammatory factors TNF-α and IL-1β, and promoted epithelial regeneration and collagen deposition	[79]
PCL/gel nanofibrous composite scaffold containing silicate-based bioceramic particles (NAGEL)	PCL, Gel	NAGEL	PCL/Gel nanofibrous composite scaffold can promote diabetic wound healing by promoting angiogenesis, collagen deposition, re-epithelialization, and inhibiting inflammation	[80]
CS/PVA/ZnO nanofibrous membranes	PVA, CS	ZnO nanoparticles	ZnO nanoparticles have bactericidal properties, and the porous structure of the fiber membrane can promote the proliferation of fibroblasts and the recruitment of macrophages, and thus accelerate wound contraction	[81]

**Table 2 pharmaceutics-14-00006-t002:** Drug-loaded electrospun fibers for collaborative therapy of diabetic wounds.

Name of Electrospun Fiber Membrane	Spinning Polymers	Active Ingredient	Mechanism of Action	Reference
Sesamol-loaded cellulose acetate (CA) /zein nanofiber membranes	CA, zein	Sesamol	Sesamol can down-regulate expression of inflammatory cytokines, such as IL-1β and TNF-α, and up-regulate expression of interleukin-6 (IL-6) (anti-inflammatory cytokines)	[83]
Bi-layered fibrin/ poly(ether)urethane scaffold loaded with PL	poly(ether)urethane	PL	PDGF and VEGF released by PL can promote collagen deposition and re-epithelialization, thus promote diabetic wound healing	[84]
PVA-connective tissue growth factor (CTGF) /PLA core/shell nanofibrous membranes	PVA, PLA	CTGF	PVA-CTGF/PLA core/shell nanofibrous membranes are conducive to the proliferation and migration of fibroblasts, keratinocytes and other cells, which are beneficial to diabetic wound healing	[85]
17BIPHE2-PCL /pluronic F127 core/shell nanofibers	PCL, pluronic F127	Antimicrobial peptide 17BIPHE2	17BIPHE2-PCL/pluronic F127 core/shell nanofibers promote wound healing by removing bacterial biofilms from diabetic wounds	[86]
Poly (acrylic acid) (PAA)/polyvinyl pyrrolidone (PVP)-CFX /PCL triple-layered nanofibrous membranes	PAA, PVP, PCL	CFX	CFX has antibacterial effects on gram-negative and gram-positive bacteria. The antibacterial activity of PAA/PVP-CFX/PCL nanofibrous membranes gives it the potential to promote diabetic wound healing	[87]
DCH-loaded PLA nanofibrous membranes	PLA	DCH	High levels of MMPs and TNF-α converting enzyme (TACE) can prevent wound healing, and DCH can inhibit the activity of MMPs and TACE, thus promoting diabetic wound healing	[88]
Dimethyloxalylglycine (DMOG)-embedded PCL fiber membranes	PCL	DMOG	DMOG is a small molecule inhibitor of non-specific prolyl hydroxylases, which can inhibit the decomposition of HIF-α, create a cellular microenvironment similar to hypoxia, thus accelerating wound healing by activating angiogenesis and fiber regeneration	[89]
Repaglinide-loaded PVA/PVP nanofibers	PVA, PVP	Repaglinide	Repaglinide-loaded PVA/PVP nanofibers can solve the problems of poor water solubility and unstable drug absorption of hypoglycemic drug repaglinide, significantly reduce blood glucose level, and promote diabetic wound healing	[90]
Bioactive antibiotics and PDGF loaded PDGF/PLGA-antibiotic core/sheath nanofibrous	PLGA	PDGF, gentamicin, vancomycin	PDGF/PLGA-antibiotic core/sheath nanofibrous promote angiogenesis and epidermal hyperplasia through the synergistic effect of PDGF and antibiotics	[91]
Gentamicin sulfate (GS) and recombinant human epidermal growth factor (rhEGF) co-loaded Eudragit RL/RS nanofibers	Eudragit RL-100 and Eudragit RS-100	GS, rhEGF	Bacterial inhibitor GS can reduce inflammation of diabetic wounds, and rhEGF can promote granulation tissue formation and angiogenesis at the wound	[92]
Monocyte chemoattractant protein-1(MCP-1) loaded polyglycolic acid (PGA)-Gel electrospun scaffold	PGA, Gel	MCP-1	MCP-1 promotes macrophages to participate in the wound healing process, thus the growth factors VEGF and PDGF secreted by macrophages can promote wound healing	[93]
Sirt1 agonist (SRT1720) loaded PLGA/collagen protein /silk membranes inoculated with embryonic artery cluster of differentiation 133+ cells (EACCs) (PCSS-EACCs)	PLGA, collagen protein, silk	SRT1720, EACCs	PCSS-EACCs can steadily release SRT1720 for 15 days, thus improving the survival rate of EACCs in a high glucose environment. The release of vascular endothelial growth factor A (VEGFA) and interleukin-8 (IL-8) from EACCs ultimately promote endothelial cell proliferation, migration, and angiogenesis	[94]
PCL/Gel-pioglitazone nanofibrous membranes	PCL, Gel	Pioglitazone	PCL/Gel-pioglitazone nanofibrous membranes reduce expression of MMP-9, IL-1β, and IL-6 to reduce wound inflammation, and upregulate expression of macrophage inflammatory protein-2 (MIP-2), TNF-α, and VEGF to promote wound healing	[95]
Hyaluronic acid (HA) /PLGA core/shell fiber loaded with EGCG	PLGA, HA	EGCG	EGCG can promote diabetic wound healing by promoting capillary formation and epithelial cell proliferation	[96]
PLA/CS nanoscaffolds containing cod liver oil	PLA, CS	Cod liver oil	Cod liver oil enhances the activity of the growth factor, promotes cell differentiation, reduces inflammation, and increase the production of IL-1, which promotes diabetic wound healing	[97]
EGF, bFGF, antimicrobial peptide LL-37 co-loaded PVA- Silk fibroin (SF) nanofiber membrane	PVA, SF	EGF, bFGF, antimicrobial peptide LL-37	EGF and bFGF can promote the proliferation of fibroblasts, keratinocytes, and endothelial cells, antimicrobial peptide LL-37 can reduce the inflammation of the wound, EGF, bFGF, and antimicrobial peptide LL-37 can cooperate to promote diabetic wound healing	[98]

## Data Availability

Not applicable.

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
