# Peer review of "Application of Electrospun Nanofiber Membrane in the Treatment of Diabetic Wounds"

_pharmaceutics, 2021, doi:10.3390/pharmaceutics14010006_

Round 1
Reviewer 1 Report
The manuscript entitled “Application of electrospun nanofiber membrane in the treatment of diabetic wounds” reports an interesting topic based on the applications of polymer electrospun fibers, nanoparticles loaded electrospun fibers, drugs loaded electrospun fibers and cell loaded electrospun fibers in the treatment of diabetic wounds. I recommend the publication of the manuscript in Pharmaceutics after major revisions. Below, I include some specific comments.
- Line 142: acronyms should be always reported.
- In the section 2.2, the comparison with other electrospun systems could be useful to strengthen the review.
- In the section 2.2, the effect of electrospinning process parameters could be briefly discussed. Then, some drawbacks of the process could be cited.
- Line 166: avoid the double use of word “spinning”.
- Figure 3: the picture should be slightly improved in terms of quality.
- Are there similar reviews already available? Which is the novelty of this manuscript?
- The future perspectives concerning this topic should be better highlighted.
- Some minor English revisions should be accomplished.
Reviewer 2 Report
1. Despite the rather high volume of data and the breadth of the publications covered, the text lacks systematicity beyond just listing the methods and results of reviewed papers. The descriptive part of the works of other authors is replete with redundant details of the processing of materials, which are not used in the further text for any purpose, which overloads the paper. Also, the text is extremely lacking in paragraph division. A tremendously negative impression is also left by the absence of links in the captions to figures borrowed from other works.
2. In the sections about the description of the electrospinning process, there is a singular description of some papers about the release of certain drugs. Are they exceptional for these processing types and one of a kind? Why are they referred to the section of direct description of production technologies? In the following text, other works of similar nature are given.
3. The name of the materials includes the concept of a membrane, while the text lacks even a basic definition of this type of material and its features. Should the used term mean any material obtained by electrospinning? In this case, the total number of reviewed sources raises serious doubts about the value of this work due to a sufficient breadth of the topic of fiber electrospinning and their application to obtain wound dressings for defects of various etiologies.
4. The review lacks any attempt to analyze the representativeness of the reviewed papers or criteria for their exclusion, despite the fact that in vivo animal testing is present in a wide range of sources. The conclusions do not carry out any real generalization and deduction of trends in the development of membrane materials for wound dressings, limiting themselves to general words and fairly well-known facts, which casts doubt on the value of the analytical work.
Reviewer 3 Report
In this study, the applications of polymer electrospun fibers, nanoparticles loaded electrospun fibers, drugs loaded electrospun fibers, and cell loaded electrospun fibers in the treatment of diabetic wounds were reviewed, in the attempt provide new ideas for the effective treatment of diabetic wounds.
The manuscript is well written and is characterized by an high-quality in-depth analysis and rich in translational news.
The manuscrit is suitable for publication after a control of english style
Reviewer 4 Report
Authors reviewer the application of electrospsun nanofiber membrane in the treatment of diabetic wounds. The authors chose a burning topic. The MS is written clearly. I suggest addressing the following comments.
- Fabrication of electrospinning membrane for antibacterial membrane should be provided.
- Are all figures cleated by authors or taken from other reports. Please confirm and cite in caption after taking copyright permission.
- Electrospinning membranes are two-dimensional structures but this is not enough to cope the actual three-dimensional wound healing process. I suggest analysis critically. For your ref, please check these refs. Carbohydrate Polymers 273, 118603, Carbohydrate Polymers 250, 116880,
- Silver particles entrapped membranes have been used widely as antibacterial patches. Authors missed reviewing those works. Journal of Industrial and Engineering Chemistry 30, 254-260
Round 2
Reviewer 1 Report
The authors revised the manuscript according to the requested modification. I now can suggest the publication in Pharmacueutics.
